# The Application of Polycaprolactone in Three-Dimensional Printing Scaffolds for Bone Tissue Engineering

**DOI:** 10.3390/polym13162754

**Published:** 2021-08-17

**Authors:** Xiangjun Yang, Yuting Wang, Ying Zhou, Junyu Chen, Qianbing Wan

**Affiliations:** 1Department of Prosthodontics, West China Hospital of Stomatology, Sichuan University, Chengdu 610041, China; yangxj199610@163.com (X.Y.); wangyuting408@163.com (Y.W.); 18683571838@163.com (Y.Z.); 2State Key Laboratory of Oral Diseases, National Clinical Research Center for Oral Diseases, West China School of Stomatology, Sichuan University, Chengdu 610041, China

**Keywords:** polycaprolactone, three-dimensional scaffolds, bone tissue engineering

## Abstract

Bone tissue engineering commonly encompasses the use of three-dimensional (3D) scaffolds to provide a suitable microenvironment for the propagation of cells to regenerate damaged tissues or organs. 3D printing technology has been extensively applied to allow direct 3D scaffolds manufacturing. Polycaprolactone (PCL) has been widely used in the fabrication of 3D scaffolds in the field of bone tissue engineering due to its advantages such as good biocompatibility, slow degradation rate, the less acidic breakdown products in comparison to other polyesters, and the potential for loadbearing applications. PCL can be blended with a variety of polymers and hydrogels to improve its properties or to introduce new PCL-based composites. This paper describes the PCL used in developing state of the art of scaffolds for bone tissue engineering. In this review, we provide an overview of the 3D printing techniques for the fabrication of PCL-based composite scaffolds and recent studies on applications in different clinical situations. For instance, PCL-based composite scaffolds were used as an implant surgical guide in dental treatment. Furthermore, future trend and potential clinical translations will be discussed.

## 1. Introduction

Bone tissue, as one of the most important organs, plays multiple roles in daily life [1]. Lots of patients are suffering from bone disease resulting from tumor resections, trauma, infections, cysts, and injuries caused by accidents. It has been reported that over four million operations using bone grafts are performed each year to treat bone defects [2]. Autogenous bone transplantation and replacement are the main traditional options for patients with bone defects [3,4]. However, the potential risks of tissue grafts including complications and secondary injuries remain a major clinical challenge [5,6]. 

To overcome this shortage, bone tissue engineering is one of the most proposing alternative methods. Bone tissue engineering focuses on the main processes including cell growth and the customized construction of human bone tissue [7,8,9]. Further, 3D printing has multiple advantages, including precise deposition, cost-effectiveness, simplicity, and cell distribution controllability [10]. The developments and applications of 3D printing have been increasing constantly over the past few years.

In the field of bone tissue regeneration, polycaprolactone (PCL) is one of the most common materials in fabricating scaffolds. PCL is a Food and Drug Administration (FDA) approved linear polyester with good biocompatibility, slow degradation rate, less acidic breakdown products in comparison to other polyesters, and has the potential for loadbearing applications [11,12]. The slow degradation of PCL allows time for bone remodeling and can also be manipulated to adjust the polymer’s biodegradation rates [13,14]. Additionally, PCL is one of the most preferred polymers for extrusion-based 3D printing due to its melting temperature of 55–60 °C [13]. It exhibits good mechanical properties with high flexibility and great elongation, conducive to the preparation of scaffolds for craniofacial bone repair [9]. However, pure PCL has no osteogenic potential to induce bone regeneration [15]. Thus, researchers combine PCL with various polyesters, inorganic substances, metal elements, or collagen to improve the properties of the scaffolds. This review discusses and summarizes recent advancements in PCL-based composite scaffolds, focusing on the fabrication and functionalization methods and their application to promote bone growth in vitro and in vivo. Further, the future trends and potential clinical translations will be discussed.

## 2. Fabrication Techniques of Three-Dimensional Printing for Bone Scaffolds

The availability of desired properties for 3D printed scaffolds relies on the printing technology that is used. Generally, 3D printing is a process of layer-by-layer fabrication using powder, liquid, or solid material substrates. Starting from the bottom and building up, each newly formed layer is triggered to adhere to the previous layer, gradually increasing the size of the construct [16]. The techniques using in 3D printing include stereolithography (SLA), selective laser sintering (SLS), and fused deposition modelling (FDM). Because of the themoplasticity of PCL, the most common technique used for 3D printing is FDM [4,16,17,18,19]. FDM uses a temperature controlled printhead to deposit thermoplastic material onto a platform in a layer-by-layer manner to build up a 3D construct. PCL begins to melt by being driven into a heated printhead, allowing thin layers to be deposited precisely and sequentially. The molten PCL cools in the air of the printing environment, allowing it to rapidly fuse together to create a scaffold [16]. However, the elevated temperatures limit the inclusion of biomolecules and hydrogels.

Multiple studies have focused on cell behaviors on PCL scaffolds the scales of which usually range from hundreds of microns to millimeters fabricated by FDM [20]. However, there is a limitation of the techniques that can be used for the fabrication of fibrous micro-environments to study cell behaviors [21]. Electrohydrodynamic (EHD) printing, also named Melt Electrospinning Writing (MEW), is a recently developed technology to overcome the above limitations [22,23]. As shown in Figure 1, the design of MEW devices combines the advantages of conventional electrospinning and 3D printing [21]. This technology allows micron to sub-micron fiber fabrication [24]. Kim [25] fabricated fibrous scaffolds with the EHD technique and demonstrated significantly high metabolic activity and mineralization of the cells cultured on the micro-fiber PCL scaffolds. The comparison of PCL scaffolds fabricated by FDM and by MEW is shown in Figure 2. The micro-fiber PCL scaffolds printed by MEW directly affected the cell adhesion morphology. SLS is another technique for PCL-based scaffold fabrication. The basic design of a SLS printer is a housing that has a powder bed, a laser, a piston to move down in the vertical direction, and a roller to spread a new layer of powder. The computer-controlled laser beam sinters the powder, and the remaining powder works as a structural support for the scaffold being constructed [26]. With laser assistance, SLS is more accurate but more expensive than FDM.

Bioprinting is another advanced technology which has aroused wide interest in recent years. Bioprinting can be used to deposit living cells and other biomaterials to build complex tissue constructs [16,27]. For the bioprinting of PCL-based composite scaffolds, researchers usually combine PCL with hydrogels that load living cells. Bioprinters have multiple print nozzles, one for PCL scaffolds printing, and the others for cell-loaded biomaterial printing simultaneously or separately. With this technology, mesenchymal stem cells (MSCs) or human umbilical vein endothelial cells (HUVECs) were usually loaded on the scaffolds to improve the vascularization of the printed structure [28,29,30,31]. The comparison of the 3D printing techniques for PCL-based scaffold fabrication is shown in Table 1.

## 3. PCL-Based Composite 3D Scaffolds

### 3.1. The Advanced Properties of PCL-Based Composite 3D Scaffolds

Numerous scaffolds produced from a variety of biomaterials have been used in the field in attempts to regenerate different tissues and organs [16]. Generally, 3D scaffolds are designed to imitate the extracellular matrix (ECM). These scaffolds are required to possess bioactive characteristics as follows and as reported [33,34]: a porous structure for the transport of nutrients, waste products, and for the communication with other cells; good biocompatibility with the controlled degradation and absorption rate of cell/tissue growth in vitro and/or in vivo; suitable surface chemistry stimulating cell ingrowth, cell attachment, and cell differentiation; and properties matching the individual clinical environments of bone defects.

Owing to its brilliant biocompatibility and easy processability, PCL has been extensively used in scaffold fabrication. However, the poor hydrophilia and low bioactivity of pure PCL systems limit their applications in the biomedical field [34]. Combining the PCL matrix with bioactive inorganic particles as fillers provides a promising way to overcome these shortcomings [34,35]. Metals, oxides, polymers, and carbon-based materials have all been applied to PCL scaffolds for property improvement [16]. A summary of recent researches and the advances in PCL-based composite scaffolds and property improvements is shown in Table 2.

The effects of different materials on the performance of composite scaffolds were also compared. Ethan [65] focused on the comparison of PCL-based scaffolds combined with tricalcium phosphate (TCP), hydroxyapatite (HA), Bio-Oss (BO), or decellularized bone matrix (DCB). They concluded that PCL-BO and PCL-DCB hybrids were superior to PCL-HA or PCL-TCP blends for bone healing applications. Marco [66] found that different diameters of hydroxyapatite blended in the printed scaffolds had distinct performance. PCL-nano-HA scaffolds showed higher levels of alkaline phosphatase activity compared to PCL-micro-HA structures. Differing from physical mixing process, Chen [67] synthesized poly(l-lactide-*co*-caprolactone-*co*-acryloyl carbonate)(poly(LLA-CL-AC)) by ring-opening polymerization in the presence of Sn(Oct)2 as a catalyst and octanol as an initiator for the first time. They found that the stiffness of the scaffolds increased after UV irradiation cross-linking.

### 3.2. The Architecture Structure of PCL-Based Composite Scaffolds

The standard approach in bone tissue engineering is to seed and grow cells on scaffolds in vitro. Typical scaffolds are 3D porous structures temporarily mimicking the natural extracellular matrix of bone [68]. Ideally, 3D scaffolds should be highly porous, have well-interconnected pore networks, and have consistent and adequate pore size for cell migration and infiltration [69]. Scaffold architecture design can significantly influence both mechanical properties and cell behaviors. The common structures designed for printed scaffolds are shown in Table 3.

A lot of researchers have focused on the outer morphology of printed scaffolds. Cylindrical and cube-shaped structures are common 3D printing shapes in preliminary studies. Similarly, circular, sinusoidal, and conventional orthogonal models were also fabricated and compared in previous studies, as shown in Figure 3 [70]. The results demonstrated that less orthogonal elements enhanced osteogenic performance. Further, the scaffold shapes are usually designed to match the shape of bone defect area for clinical application.

Numbers of studies have examined the inner structure inducement on cells behavior. First, the different deposit angle, which usually has an effect on mechanical property and porosity of scaffolds, has been studied [16]. The scaffold porosity is an important factor affecting the performance of scaffolds. Shim [71] found that 3D printed PCL GBR membranes with 30% porosity (130 μm pore size) were excellent for calvarial regeneration. Second, the pore structure has also been studied. Yang [72] developed compatible scaffolds which included macropores, medium-sized pores, and small pores, and these scaffolds are tailored to be similar to that of natural extracellular matrix (ECM). Adeola [73] focused on the effects of pore geometry on modulating mechanical behavior of PCL scaffolds. Lee [74] found that the kagome structure obviously improved the mechanical properties of PCL scaffolds compared to the grid structure. Abigail [75] fabricated artificial models that mimic the microstructure of bone, improving the accuracy of bone grafts.

In the field of tissue engineering, personalized medicine highlights the use of specifically designed scaffolds. The customized scaffolds can optimize the repairing process in cases of irregular-shaped wounds and tissue defects, especially for orthopedic, oral, and maxillofacial surgery [37]. With the aid of computed tomography (CT) or magnetic resonance imaging (MRI) data, the fabrication of patient-customized scaffolds using 3D printing technology is realizable. In this way, the mechanical properties, pore size, and porosity of the scaffold can be controlled [88]. Bae [89] successfully conducted the implant process using PCL-based scaffolds in the beagle model (Figure 4). These scaffolds restored the original volume and shape of the alveolar ridge in the defect site and performed well as a surgical guide to place the implant at the proper location and depth.

### 3.3. Cell-Laden PCL-Based Composite Scaffolds

Naturally based hydrogels can offer excellent cellular interaction and biocompatibility, but they suffer from poor mechanical properties. On the other hand, PCL printed scaffolds possess compressive strengths within the range of cortical bone [90,91]. This has aroused great interest in the 3D bioprinting of cell laden hydrogel bioinks reinforced with stiffer PCL fibers [92,93,94]. Combining these two materials, cells can be diffused on the scaffolds accurately with the bioprinting process. Researchers demonstrated that the cells which were injected into the pores of scaffolds before clinical implantation, indicating promising osteogenesis enhancement in vivo [88].

There are a number of hydrogels that can be used for cell loading medium, such as matrigel [95], alginate [30,96], agarose [92], hyaluronic acid [78,92], and GelMA [31,92]. In the study of Caroline Murphy [95], they presented a scaffold with a mixture extrudable paste of PCL and borate glass. Human adipose stem cells suspended in matrigel were then ejected inside of the scaffold as droplets. They found a controlled release of the bioactive glass for up to 14 days with degradation of the scaffolds. The results showed a high level of angiogenesis in the interior the scaffold. Stichler [78] prepared hyaluronic acid/poly(glycolic acid) mixed hydrogels using UV light cross-linking. After loading human and equine mesenchymal stem cells, the PCL-based composite scaffolds were prepared using double-nozzle 3D printing technology. The cells showed a good chondrogenic differentiation prospect after 21 days. The combination of PCL and hydrogel improves the mechanical properties. At the same time, the existence of hydrogel is more suitable for the growth and reproduction of cells than pure PCL. Due to the high plasticity of PCL and the advantages of 3D printing technology, construction of personalized biomimetic tissue has become practical.

It has been well-established that lack of vascularization within the engineered bone grafts is a major barrier to bone healing [97,98]. Aiming to overcome this problem, Mitchell [28] presented a hydrogel-based prevascularization strategy to generate prevascularized bone scaffolds. They coated co-culture PCL/HA scaffolds with hydrogels, which encapsulated ADMSC and HUVEC. This co-culture system promoted vascularization in vitro and in vivo. Similarly, Wen [29] fabricated a PCL/polydopamine-modified calcium silicate scaffold loading with ADMSC and HUVEC. Xie [30] incorporated mesenchymal stem cell-derived microvesicles into alginate/ PCL constructs for angiogenesis promotion.

### 3.4. Carrier Function of PCL-Based Composite Scaffolds

Scaffold bioactivity can be increased by adding components that are able to interact with or bind to living tissues [99]. For a better clinical effect, PCL-based scaffolds with therapeutic agents added during the 3D process can be used as a carrier to load drugs and other bioactive substances to realize a long-term release.

Multiple studies have investigated the pharmacokinetic of 3D printed scaffolds. Hoang [100] used porogen leaching and 3D printing techniques and created microporous PCL scaffolds with micropors for drug loading and releasing control. They found that microscale porosity avoided the burst release of drugs and maintained relatively long-lasting drug concentrations. A summary of drug loading in PCL-based composite scaffolds is presented in Table 4.

Biomaterials are designed to release bioactive substances at the injury site to stimulate bone repair [1,105]. The modification of bioactive components on 3D scaffolds, which enable bone cells to function in a sustainable manner, has aroused great interest worldwide. To mimic the physiological bone hierarchy, the bioactive molecules were added to PCL-based scaffolds for osseointegration (Table 5).

Bone morphogenetic proteins (BMPs) are considered to be the most eminent in advancing bone growth by inducing osteogenic differentiation [1,6,15]. In Jang’s study [106], they made alginate/BMP-2/umbilical cord serum (UCS) coated on 3D-printed PCL scaffolds and demonstrated that the simultaneous use of low-dose BMP-2 and UCS significantly increased osteogenesis based in vitro and in vivo. To promote successful bone regeneration, efficient vascularization is a pre-requisite. Therefore, the angiogenic growth factor VEGF and its controlled delivery play a vital part in bone regeneration [107]. Additionally, Eric [108] constructed PCL scaffolds seeded with microspheres containing VEGF and VEGF with either BMP-2 or FGF-2 and observed significantly higher vascular ingrowth and vessel penetration than the controls. Collagen type I (COLI) can also be coated to the 3D printed structure, promoting the proliferation of chondrocytes [109]. Moreover, Won [110] compared the ability of promoting cell activity and mineralization between rhBM-2 and platelet-rich plasma (PRP) in scaffolds and concluded that rhBMP-2 was more efficient. As for mental molecule modification, gold nanoparticles (GNPs) grown on the polydopamine (PDA) coating of scaffolds have been demonstrated to be effective for bone regeneration [111]. Further, Michal [112] confirmed that osteoclast activity was greatly suppressed by the lithium release of printed PCL scaffolds.

## 4. PCL-Based Composite Scaffolds Utilized in Different Situations

For clinical application, three types of clinical applications for 3D-printed technologies are defined in previous studies [125]. The first is for prosthetic rehabilitation (improve patient aesthetic), the second is for reconstruction (tissue grafting), and last is for tissue regeneration (recapitulate native tissue structure and function). PCL-based scaffolds can be used in many kinds of tissue engineering, such as skin regeneration [126], skeletal muscle tissue regeneration [127], and tendon regeneration [73]. In the field of bone defect therapy, there are reconstruction and regeneration processes for cartilage tissue and bone tissue.

### 4.1. Reconstruction and Regeneration of Cartilage Tissue Using PCL-Based Composite Scaffolds

PCL is reported as a popular material in cartilage regeneration. Combining PCL with agarose and GelMA or MECM, engineered meniscus were constructed, which had the potential for use as a substitute for total meniscus replacement [10,128]. Zhang [129] and his colleagues constructed scaffolds for total meniscal substitution in a rabbit model. With the aid of multidetector CT and computer design, 3D scaffolds for use in total ear reconstruction were successfully fabricated [130]. Additionally, 3D printing techniques were also used for trachea engineering. For instance, Parket [131], Shan [86], and Gao [132] have successfully fabricated trachea scaffolds. Interestingly, Parket implanted the tracheal scaffold into the omentum before tracheal scaffold implantation in rabbits and concluded that the omentum-culture of the tracheal scaffold was beneficial for rapid the re-epithelialization and revascularization of the scaffold. Further, it also prevented postoperative luminal stenosis.

### 4.2. Reconstruction and Regeneration of Bone Tissue Using PCL-Based Composite Scaffolds

For animal surgery, Carla [133] conducted a surgical therapy of a chronic oronasal fistula in a cat using autologous platelet-rich fibrin and bone marrow loaded printed PCL scaffold. A CT scan revealed complete healing after a six-month follow-up. In Lee’s research [87], a customized scaffold matched with an 8-shaped bone defect on the rabbit calvarium model was designed according to 3D computed tomography. they then implanted the scaffold in the defect area, which showed excellent mechanical robustness and enhanced osteoconductivity. Rebecca Chung [134] proposed a patient-specific 3D printed bioresorbable graft substitute for segmental bone replacement.

In the dental application field, it is difficult to fabricate a scaffold matching the complex shape and functions of the nature craniomaxillofacial (CMF) bones. With the CT multidetector data, we can construct patient-customized structure 3D scaffolds for bone tissue regeneration using 3D printing [16,18]. Joshua printed an anatomically shaped scaffold that closely resembled the 3D models [135]. A custom scaffold can be used simultaneously as an implant surgical guide and as a bone graft in a large bone defect site. Upon dental implant surgery, successful implant placement is reliant on adequate alveolar bone volume at the implant site, which can provide mechanical stability for dental implants. It is important to augment the alveolar ridge for enabling the placement of dental implants and thus to restore both functionality and esthetic appearance [90]. Rider [90] and Vaquette [136] proposed that printed scaffolds showed potential in transferring to alveolar vertical bone augmentation (Figure 5). Due to the high compressive strength of the printed structure, these scaffolds may be applicable for procedures involving simultaneous implant placement and ridge augmentation.

## 5. Conclusions

PCL is a common polymer with unique biomedical and mechanical properties that make it favorable for a wide range of bone tissue engineering applications. Its low degeneration allows the imperative periods needed for new bone regeneration. With the development of three-dimensional printing techniques, various 3D structures can be fabricated successfully. Numbers of materials were utilized in the studies on PCL-based composite scaffolds, and composite scaffolds demonstrated superior performance to pure PCL scaffolds in recent studies. Cells can be also printed into the scaffolds by blending with hydrogel, which provides a compatible medium for cell proliferation. Further, in vivo studies of PCL scaffolds used in bone or cartilage tissue engineering applications have proven their osteogenic potential. Nevertheless, these studies were mostly conducted in small animals (usually rats and rabbits), which may not sufficiently predict clinical application in humans. Almost none of the researchers have proceeded to the phase of human trials yet. Thus, the advantages of PCL-based tissue engineering remain distant for patients in hospital. However, PCL is still a promising biomaterial. Future work should focus on PCL-based scaffolds in large animal models as well as in human clinical trials. We expect to develop custom-made 3D composite scaffolds that can be grafted directly with stem cells in clinical practice.

## Figures and Tables

**Figure 1 polymers-13-02754-f001:**
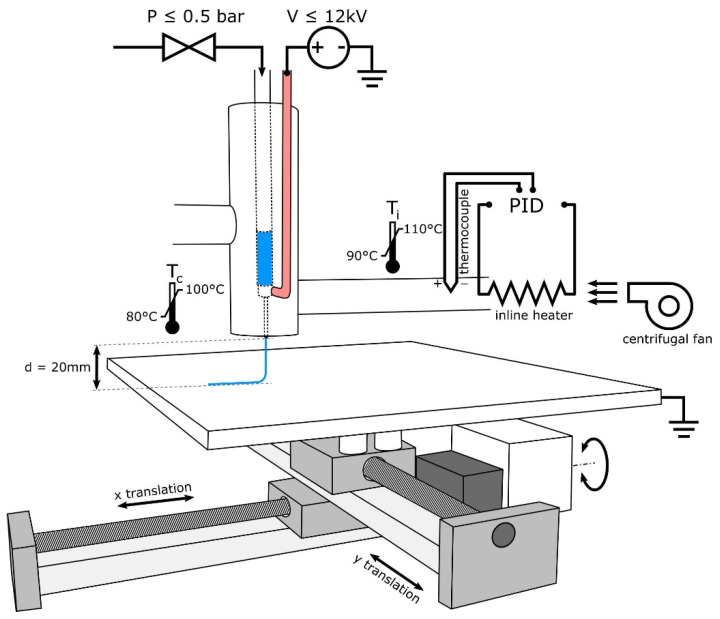
Schematic of MEW device design illustrating the use of heated air to control syringe and needle temperature while using air pressure and high voltage to draw PCL and to produce electrospun fibers [21]. Copyright 2018 Elsevier.

**Figure 2 polymers-13-02754-f002:**
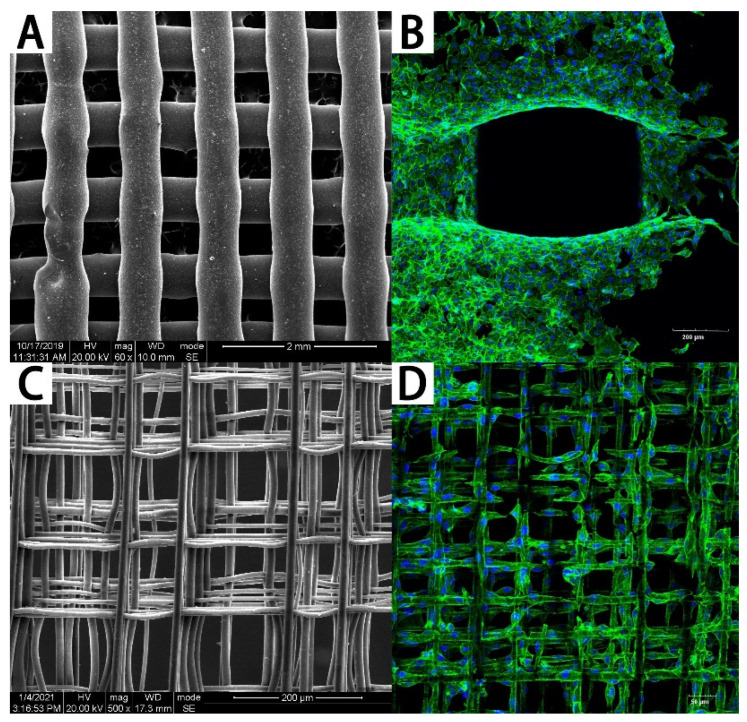
The PCL scaffolds fabricated by FDM and MEW. (**A**) The PCL scaffold fabricated by FDM and (**B**) cells cultured on it. (**C**) The PCL scaffold fabricated by MEW and (**D**) cells cultured on it.

**Figure 3 polymers-13-02754-f003:**
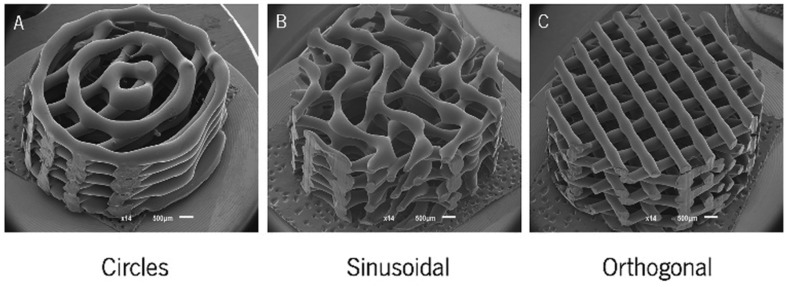
SEM micrographs of the (**A**) circle, (**B**) sinusoidal, and (**C**) orthogonal scaffolds produced [70]. Copyright 2018 MDPI.

**Figure 4 polymers-13-02754-f004:**
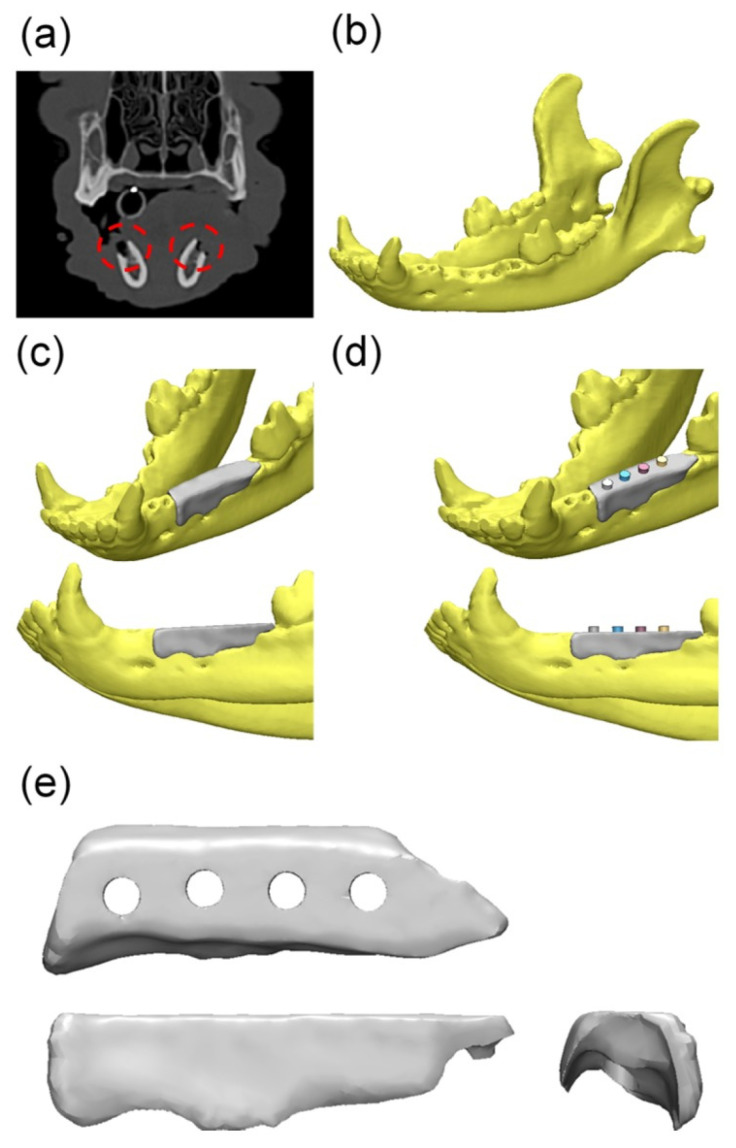
Overall modeling process of the implant guide scaffold: (**a**) Red dashed line: alveolar bone defect of mandible; (**b**) 3D modeling process of CT image; (**c**) 3D scaffold cover of the defect area; (**d**) 4 thru holes for inserting implant fixture; (**e**) Final model of implant-guided scaffold [89]. Copyright 2017 MDPI.

**Figure 5 polymers-13-02754-f005:**
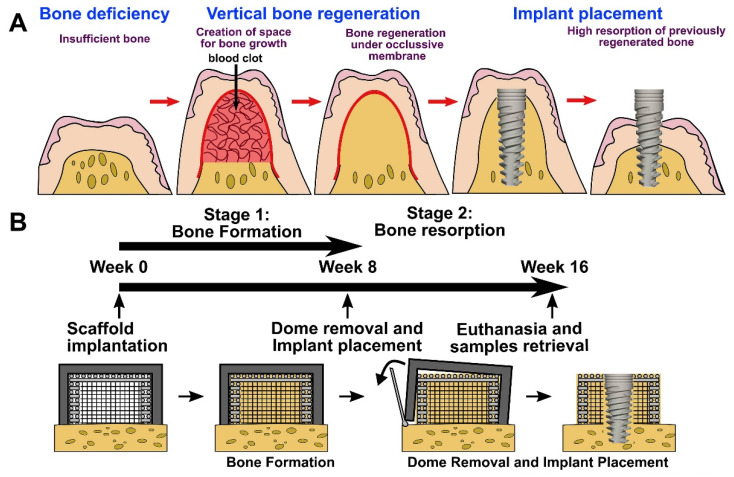
Vertical bone augmentation with a 3D printing approach. (**A**) Description of the clinical problem following surgical re-entry in previously elevated bone, resulting in significant bone resorption. (**B**) Timeline of the experimental approach involving a two-staged strategy; bone formation following surgical re-entry and implant placement [136]. Copyright 2021 Elsevier.

**Table 1 polymers-13-02754-t001:** Comparison of 3D printing technique for PCL-based scaffold fabrication.

Technique	Costs	Cell Loading	Advantages	Potential Disadvantages	References
FDM	Low	No	High mechanical strengths; simple process; no supports needed; no solvent required; high production rate	High temperatures; limited bioactivity	[16,17,18,26,32]
SLS	High	No	High resolution; Fast processing; no supports needed;	Thermal damage; rough surface finish	
EHD	medium	No	Creates micro-fibers for cell attachment;	Organic solvents may be needed; poor mechanical properties	
Bioprinting	medium	Yes	Good bioactivity; cells and hydrogels can be printed	Low accuracy; costs can be expensive for machinery	

**Table 2 polymers-13-02754-t002:** A summary of recent research and advances in PCL-based composite scaffolds and property improvement.

Researcher	Materials	Procedure	Properties Improved	Encapsulated Cell	Preclinical Study	Achievements
Seyedsalehi [36]	PCL/reduced GO	Solvent evaporation film casting	Mechanical properties; biocompatibility	Human adipose-derived stem cells (hADSCs)	--	Significantly improving the compressive strength and stiffness
Liu [37]	PCL/HA/VEGF	Emulsification; solvent evaporation; surface modification	Vascularization	rMSCs	Rat cranial defect model	Enhancing the vascularized bone regeneration
Wu [38]	PCL/CS/dECM	Melt blending; coating	Hydrophilia; biocompatibility	Human Wharton’s Jelly; mesenchymal stem cells	--	Excellent biocompatibility, anti-inflammatory characteristics
Wang [39]	PCL/graphene	Melt blending	Biocompatibility; osteogenesis	Human adipose-derived stem cells (hADSCs)	Rat; calvaria critical size defect	New tissue formation, well-organized tissue deposition and bone remodeling
Janitha [40]	PCL/GO	Mix solution	Osteogenic capability; mineralization	Urine; preosteoblast cell line (OB6)	--	Increased cell attachment and proliferation; increased mineralization
Park [41]	PCL/GO	Coating	Biocompatibility; osteogenesis	Periodontal ligament stem cells (PDLSCs)	--	Promoting the cell proliferation and osteogenic differentiation
Huang [42]	PCL/CNT	Melt blending	Biocompatibility; mechanical properties	hADSCs	--	Enhancing protein adsorption, mechanical, and biological properties
Julia [43]	PCL/nHAP	Coating layer	Osteoinductivity; osteoconductivity	Human osteosarcoma cell line MG-63	Rabbits; a 5 mm round hole on the iliac crest tuber sacrale	Strongly stimulated new bone tissue formation
Onur [44]	PCL/nHA/cMgF2	Melt blending	Mechanical properties; osteoinductivity;	Human fetal mesenchymal stem cells (MSCs)	--	Increasing stiffness and toughness; better performance of osteogenic differentiation and stimulated mineralization
Petretta [45]	PCL/Mg-Containing Bioactive Glasses	Mixed solution	Mechanical properties; biocompatibility	Human bone-marrow-derived mesenchymal stem cells (BM-MSCs)	--	High level of biocompatibility, bioactivity, and cell adhesion
Sławomir [46]	PCL/TCP	Twin-screws extruder injection molding	Mechanical properties; osteoinductivity	hADSCs	--	3D culture promoted cells proliferation
Luo [47]	PCL/OSP	Melt blending	Crystallinity properties; mineralization ability	MG-63 cells	--	The scaffolds showed a strong ALP activity
Kazim [48]	PCL/PLGA/nHA	Homogenized solution	Mechanical properties; osteoconductivity	Primary culture rat bone marrow stem cells (rBMSCs)	Rat calvarial defects	Promoted cell attachment and proliferation; faster Degradation; newly formed mineralized tissue
Alexandra [49] Hung [50]	PCL/DCB	Melt blending	Osteogenic capability	ASCs	--	Great osteoinductivity of the scaffolds
Su [51]	PCL/PEG	Heating blending	Biocompatibility; wetability	MG-63	--	Increased hydrophilicity; improved cellular proliferation
Kim [52]	PCL/TCP/dECM	Melt blending; immersing	Biocompatibility; Osteogenesis	Preosteoblastic MC3T3-E1	Rabbit calvarial defect	Excellent cell seeding efficiency, proliferation; outstanding results of bone regeneration
Joseph [53]	TCP/PCL	Co-deposition or coating	Mechanical properties;	--	--	Improved flexural strength, flexural modulus, and fracture toughness
Elnaz [54]	PCL/BG particles	Coating	Mechanical strength; bioactivity	Preosteoblastic mouse calvaria cells (MC3T3-E1)	--	Improved stiffness; more hydrophilic nature;more porosity; better cell attachment and proliferation
Meik [55]	PCL/PCL/Ca-polyP-MP	Melt blending	Mechanical strength; morphogenetic activity.	Primary human osteogenic sarcoma cells (SaOS-2 cells)	--	Attracting and promoting the growth of human bone-related SaOS-2 cell
Hwang [56]	PCL/PLGA/β-TCP	Melt blending	Biocompatibility; osteogenesis	--	Rats calvarial defect	Better ability to maintain bone defects and to support barrier membranes
Park [57]	PCL/β-TCP	Dry-mixed	Biocompatibility; osteogenesis	D1 mouse mesenchymal stem cell lines	--	Increased the surface roughness, porosity, and the wettability, and effectively promoted cell growth and osteogenic differentiation
Shim [58]	PCL/BCP	Surface immobilized; mixed	osteogenesis	MG-63	Rats tibial defect model	Increased new bone formation and mineralized bone tissues
Chiu [59]	PCL/MTA	Thermal pressing	Mechanical strength; osteogenesis	Human dental pulp cells (hDPCs)	--	Effectively promoted the adhesion, proliferation, and differentiation of hDPCs; increasing compression strength
Donata [60]	PCL/PEDOT	Vapor-phase polymerization	Wettability	hfMSCs	--	Increased surface roughness and wettability
Miao [61]	PCLtroil/castor oil	Mixed	Biocompatibility	MSC	--	Excellent attachment, proliferation, and differentiation of MSCs
Elsa [62]	PCL/nHA/CNT	Mixed solution	Electrical conductivity; biocompatibility	MG63	--	Typical hydroxyapatite bioactivity, good cell adhesion, and spreading at the scaffold surface
Pedram [63]	PCL/nHA/CNW	Melt blending	Biological and mechanical properties	MC3T3-E1	--	Significantly increasing the biological and mechanical properties
Wang [64]	PCL/nHA/CaO_2_/gelatin	Melt blending; coating	Biocompatibility; osteogenesis	BMSCs	New Zealand white rabbits; the osteonecrosis of femoral head	Enhancing the angiogenesis and survival of grafted stem cells

ABBREVIATIONS: calcium silicate (CS); polycaprolactone (PCL) decellularized extracellular matrix; graphene oxide (GO); carbon nanotubes (MWCNT); hydroxyapatite nano powder (nHA); magnesium fluoride nanoparticle (cMgF2); tricalcium phosphate (TCP);oyster shell powder (OSP); poly(d,l-lactide-*co*-glycolide) (PLGA); decellularized bone matrix (DCB); polyethylene glycol (PEG); bioactive glass(BG); calcium-polyphosphate microparticles(Ca-polyP-MP); biphasic calcium phosphate(BCP); polymer poly(3,4-ethylenedioxythiophene)(PEDOT); Chitin–Nano–Whisker (CNW).

**Table 3 polymers-13-02754-t003:** The structure design of printed scaffolds.

Inner Structure	Types	3D Printing Technique	References
Orthogonality	Equal patterning	FDM	[21,59,70,73,76,77,78]
	Gradient patterning	FDM	[20,61]
Oblique crossing	0/45°/90°/135° laydown pattern	FDM	[55,73,79]
	0/60°/120° laydown pattern	FDM	[76,80]
	0/30°/60°/90°/120°/150° laydown pattern	FDM	[76]
	0/15°/30°/45°	FDM	[76]
	0/30°, 0/60° microfiber angel	EHD	[21]
	0/45°/90° laydown pattern	FDM	[81]
	0/45° laydown pattern	FDM	[73]
	Irregular	FDM	[21]
spiral-like struts	--	FDM	[82,83]
Circle	--	FDM	[70]
Sinusoidal	--	FDM	[70]
Irregular cribrate	--	FDM	[84]
Surface porous	--	FDM	[47,85,86]
Kagome structure	--	FDM	[74,87]
Honeycomb-like	--	FDM	[81]

**Table 4 polymers-13-02754-t004:** A summary of drug loading in PCL-based composite scaffolds.

Drug	Pesticide Effect	Duration	References
sodium indomethacin	Anti-inflammation; analgesia	8 h with an 83.36% (±1.88) drug releasing	[101]
Lidocaine	Pain relief;	4–7 days	[102]
Silver nanoparticles	antimicrobial;	80% degradation in 20 days;	[9]
Ag_3_PO_4_	preventing infections	3% loaded for at least 7 days;	[102]
Alendronate	Induced the osteogenic differentiation of osteoblasts	Slow release as a result of slow degradation of the PCL polymers	[103]
Levofloxacin	Anti-inflammation	A fast release in the first few days and a sustained release up to 5 weeks	[104]

**Table 5 polymers-13-02754-t005:** Bioactive molecule inclusion in PCL-based scaffolds for osseointegration.

Bioactive Molecule	Medium	References
BMP-7	Hyaluronic acid	[113]
Bone marrow clots	Blood	[79]
rhBMP-2	Alginate; bdECM; HA/TCP; polydopamine	[106,108] [89] [110] [114]
Lyosecretome	Alginate-based hydrogel	[115]
Interleukin-4	GelMA	[116]
Platelet-rich plasma (PRP)	-- HA/TCP	[117] [110]
plasmid DNA	Alginate and nano-hydroxyapatite	[118]
VEGF	Poly (lactic-*co*-glycolic acid)	[108]
FGF-2	Poly (lactic-*co*-glycolic acid)	[108]
Insulin-like growth factor-1 (IGF-1)	PLGA nanoparticles	[119]
Collagen type I	--	[109]
Arg-Gly-Asp	Alginate	[60]
Gold nanoparticles	polydopamine	[111]
Lithium	--	[112]
Borate glass	--	[95]
Bio-Oss	--	[65]
ZnO	nHA	[120]
Mg	--	[121]
MgF_2_	--	[44,122]
Sr^2+^/Fe^3+^	Nano-hydroxyapatite	[123]
Strontium	SrO	[124]

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
