# Peer review of "The Application of Polycaprolactone in Three-Dimensional Printing Scaffolds for Bone Tissue Engineering"

_polymers, 2021, doi:10.3390/polym13162754_

Round 1

Reviewer 1 Report

Manuscript: The application of polycaprolactone in three-dimensional printing scaffolds for bone tissue engineering

The manuscript presents very good work related to polycaprolactone in three-dimensional printing and going to be interesting for the readers.

Some minor comments are as follows.

  1. Authors need to include some interesting data in the abstract part of the manuscript.
  2. English must be improved.
  3. Novelty of the work be established.
  4. All the important results reported be compared in a tabular form to establish the superiority of the work.
  5. Authors need to add future prospective of presented research in the conclusion part of the manuscript.
  6. Authors need to incorporate some recent reference related to the manuscript to make it more interesting for the readers.

Reviewer 2 Report

The manuscript needs revision as it has some drawbacks.  only the tables seem good and most references are updated. my detailed comments are listed below. they should be carefully implemented to inmprove the manuscript

1-sect 4 has subsections which are not in same goal. please consider better classification. Also they are too short subsections.

2-The order of materials in tables are not good. for instance, table 4: would be better to bring cells, then other non-live materials, e.g. Lithium

3-New ref should be added in the introduction, or would be better in section 2 e.g., Mat Sci & Eng: C 2021,  124, 112057; Acta Biomaterialia 2021, 122, 1, Pages 26-49 ; Sec 2 needs revision and does not compare different method, e.g., FDM , bioprinting, etc. Also it would be good to have a scheme to show different method of printing, if possible

4-Please add multicompartment figures including scheme+ figure data (curve and qualitative image). Please find very good and eye-catching figures for your section. Inside of the figures, the please do not use Time new Roman font. Please use Helvetic (Roboto) font. At  least 4 figures should be added. Please see and use and cite J. Med. Chem. 2020, 63, 15, 8003–8024; 

5-References: There are some old references. Please remove/replace the outdated references. Some of them are listed below but please replace the other references.

 Fedorovich NE, Alblas J, Hennink WE, Oner FC, Dhert WJ. Organ printing: the future of bone regeneration? [J]. Trends 304
Biotechnol.2011,29(12):601-606.

6- I haven't seen application in dentistry in regenerative medicine. please add; see the above suggested paper.

7-perspective should be added
